# Distinct Cellular Tools of Mild Hyperthermia-Induced Acquired Stress Tolerance in Chinese Hamster Ovary Cells

**DOI:** 10.3390/biomedicines10051172

**Published:** 2022-05-19

**Authors:** Ádám Tiszlavicz, Imre Gombos, Mária Péter, Zoltán Hegedűs, Ákos Hunya, Barbara Dukic, István Nagy, Begüm Peksel, Gábor Balogh, Ibolya Horváth, László Vígh, Zsolt Török

**Affiliations:** 1Institute of Biochemistry, Biological Research Centre, 6726 Szeged, Hungary; tiszlavicz.adam@brc.hu (Á.T.); gombos.imre@brc.hu (I.G.); peter.maria@brc.hu (M.P.); hunya.akos@brc.hu (Á.H.); dukic.barbara@brc.hu (B.D.); bpeksel@gmail.com (B.P.); balogh.gabor@brc.hu (G.B.); hibi@lipidart.com (I.H.); vigh.laszlo@brc.hu (L.V.); 2Core Facilities, Biological Research Centre, 6726 Szeged, Hungary; hegedus.zoltan@brc.hu (Z.H.); nagy.istvan@brc.hu (I.N.); 3Department of Biochemistry and Medical Chemistry, Medical School, University of Pécs, 7624 Pécs, Hungary; 4Seqomics Biotechnology Ltd., 6782 Mórahalom, Hungary

**Keywords:** stress, heat shock response, unfolded protein response, membrane, lipidomics, membrane lipid metabolism, transcriptomics, acquired stress tolerance, Chinese hamster ovary cells

## Abstract

Mild stress could help cells to survive more severe environmental or pathophysiological conditions. In the current study, we investigated the cellular mechanisms which contribute to the development of stress tolerance upon a prolonged (0–12 h) fever-like (40 °C) or a moderate (42.5 °C) hyperthermia in mammalian Chinese Hamster Ovary (CHO) cells. Our results indicate that mild heat triggers a distinct, dose-dependent remodeling of the cellular lipidome followed by the expression of heat shock proteins only at higher heat dosages. A significant elevation in the relative concentration of saturated membrane lipid species and specific lysophosphatidylinositol and sphingolipid species suggests prompt membrane microdomain reorganization and an overall membrane rigidification in response to the fluidizing heat in a time-dependent manner. RNAseq experiments reveal that mild heat initiates endoplasmic reticulum stress-related signaling cascades resulting in lipid rearrangement and ultimately in an elevated resistance against membrane fluidization by benzyl alcohol. To protect cells against lethal, protein-denaturing high temperatures, the classical heat shock protein response was required. The different layers of stress response elicited by different heat dosages highlight the capability of cells to utilize multiple tools to gain resistance against or to survive lethal stress conditions.

## 1. Introduction

For survival, living beings are required to maintain a complex dynamic equilibrium, also called homeostasis, to cope with internal and external challenges. Under adequate circumstances, the growth and development of organisms are enhanced, yet, a proper response is inevitable upon suboptimal changes in the individual’s environment [1,2]. The disharmony by these changes, however, may be either beneficial (eustress) or harmful (distress) [3]. Consequences of stress depend on the dose of stress, such as intensity and duration. One of the most studied stress types is hyperthermia. Based on the course of the cellular stress response, we have previously classified heat stress (HS) into three distinct categories, namely: severe (involving heat shock protein (Hsp) induction and major macromolecular damage or even cell death), moderate (negligible protein denaturation, less intense Hsp induction) and mild (eliciting eustress without Hsp induction) [4]. All these conditions are characterized by the appearance of thermotolerance, which was previously associated merely with the synthesis and accumulation of the molecular chaperone Hsps (especially Hsp70 and Hsp25/27) [5]. Tolerance to stress may develop as a consequence of the heat shock response (HSR), however, other protective pathways, e.g., the unfolded protein response (UPR), might also play a role during mild heat stress [6,7].

The UPR was originally associated with the presence of unfolded proteins [8], however, a number of studies link it with perturbations of cellular membranes [9,10,11,12,13,14]. As mild heat is capable of altering the membranes, thus potentially generating lipid bilayer stress, the UPR might be initiated to restore homeostasis [6]. The UPR, which is often referred to as an endoplasmic reticulum stress response [15,16], involves the activation of three signaling branches classified by their ER-localized transmembrane sensors, i.e., inositol-requiring enzyme 1 (IRE1), protein kinase RNA-like ER kinase (PERK) and activating transcription factor 6 (ATF6) [17]. As claimed by a recent review, IRE1 and PERK also participate in the sensing of generalized lipid bilayer stress at the ER, while activation of ATF6 is triggered by an increase in specific sphingolipids [11].

Apart from protein denaturation, RNA structural rearrangements or a change in redox state, HS could also be sensed at the membrane level, especially under mild stress conditions [18,19]. Based on the “membrane sensor” hypothesis, mild heat shock, such as fever, triggers subtle changes in the fluidity and/or organization of cell membranes, which results in fine temperature sensing and initiates signaling for cellular response to stress [20].

Fever, an evolutionary version of mild hyperthermia (38–41.5 °C) and associated with many diseases [21,22], could also be regarded as a non-proteotoxic HS [4,23,24] with important therapeutic value. The broad applicability of heat therapy in curing diseases, e.g., as an enhancer of radiotherapy or anti-cancer drugs, makes it important to understand the underlying molecular details [4,25]. Similarly to heat, membrane perturbing agents—e.g., benzyl alcohol (BA)—could also be applied to perturb membrane structure and increase its fluidity, highlighting the possibility of applying adjuvants to fine-tune hyperthermic therapies [26,27,28].

We found earlier that cells preconditioned (or primed) with a short mild heat treatment (40 °C for 20 min) acquire thermotolerance in the absence of the induction of the two main chaperone proteins (Hsp70 and Hsp25) [4]. We suggested that the observed thermotolerance originated from plasma membrane microdomain/raft reorganization, membrane stabilization and heat-elicited lipid remodeling. In the case of Chinese hamster ovary (CHO) cells, an increase in the amount of ceramides, long chain fatty acid-containing sphingolipid species, cardiolipin, phosphatidylglycerol, phosphatidylserine and the depletion of lysophosphatidylcholines could be detected [4].

In the current study, we dissected how the response of CHO cells to prolonged mild hyperthermia (0 to 12 h) contributes to the development of distinct stress resistance with or without the classical Hsp response. Our results indicate that mild heat triggers a dose-dependent remodeling of the lipid bilayer structure, making it more resistant towards membrane perturbations modeled by a large dosage of the membrane fluidizer BA.

## 2. Materials and Methods

### 2.1. Cell Culture

The CHO (CCL-61, ATCC) cell line was obtained from the American Type Culture Collection and was cultured in Ham’s F-12K nutrient mix supplemented with 10% heat-inactivated fetal bovine serum (Gibco, Waltham, MA, USA) and 1% penicillin/streptomycin (Sigma-Aldrich, Burlington, MA, USA) and grown at 37 °C in a humidified 5% CO_2_ atmosphere. The cells were seeded at 250 cells/mm^2^ and were grown in tissue culture plates (VWR, Radnor, PA, USA) 24 h prior to experiments, reaching about 80–90% confluency to ensure logarithmic growth during treatments. Fresh medium was added to cells upon passaging every 2 days to maintain cell health.

### 2.2. Western Blotting

The expression levels of Hsps were probed with specific antibodies in CHO cells. After heat treatments, cells were collected and lysed with ice cold radioimmunoprecipitation assay buffer supplemented with protease inhibitor cocktail (Abcam, Cambridge, UK). Total protein content was measured using a BCA™ Protein Assay Kit (Thermo Scientific, Waltham, MA, USA) and equal amounts were loaded and separated on 12% SDS gels (for all other determinations). The gels were blotted onto polyvinylidene difluoride (PVDF) membranes. Membranes were blocked with blocking buffer containing PBS, 0.1% Tween-20 (Sigma-Aldrich) and 5% dried skim milk for 30 min at room temperature and probed with primary antibodies that recognized Hsp25 (ADI-SPA-801, Enzo Life Sciences, Farmingdale, NY, USA), Hsp70 (ADI-SPA-810, Enzo) at 4 °C overnight. Membranes were then incubated with peroxidase conjugated secondary antibodies (A3682, anti-mouse and A9169, anti-rabbit, Sigma-Aldrich) for 1 h at room temperature. Subsequently immunoreactive proteins were visualized with Immobilon Western Chemiluminescent HRP Substrate (Merck-Millipore, Burlington, MA, USA). Enhanced chemiluminescence was detected and analyzed using AlphaView-FluorChem FC3 (Cell Biosciences, Santa Clara, CA, USA). For protein loading control and data normalization, anti-GAPDH (G9545, Sigma-Aldrich) antibody was used. Primary and secondary antibody dilutions were made according to the manufacturers’ recommendations. Hsp25 and Hsp70 primary antibodies were diluted as 1:1000 while GAPDH was prepared as 1:20,000 and dilution for secondary anti-mouse and anti-rabbit antibodies was 1:160,000.

### 2.3. Cell Survival Protocol

CHO cells were submitted to stress twice. The first stress (heat pretreatment or priming) involved incubation of cells for various durations (0 to 12 h) at different temperatures (40 °C and 42.5 °C) in an atmosphere of 5% CO_2_. After priming, cells were directly exposed to a secondary sublethal stress (challenging stress). These were HS (46 °C, 20 min), membrane fluidity stress (95 mM benzyl alcohol (Sigma-Aldrich) for 20 min at room temperature) and oxidative stress (250 µM *tert*-butyl hydroperoxide (Sigma-Aldrich) for 3 h at 37 °C). 

Immediately after secondary stress, live cells were trypsinized and counted with 0.4% Trypan Blue Stain (Invitrogen, Waltham, MA, USA). Then, equal numbers of cells were plated for colony formation. Colonies were formed after 6 to 8 days, which was followed by cell fixation with methanol and staining with crystal violet (Reanal, Budapest, HU). Colonies were counted manually, and the surviving fraction was calculated (CFUtreated/CFUnon-primed control).

### 2.4. Determination of ATP Content

Cellular ATP levels were determined after heat treatments by luminescence-based CellTiter-Glo 2.0 Luminescent Cell Viability Assay (Promega, Madison, WI, USA) according to the manufacturer’s instruction. Briefly, in a 96-well plate approximately 1  ×  10^4^ cells/well were plated, then heat treated, followed by addition of equal volumes of assay reagent. Luminescence was measured using Fluoroskan Ascent FL (Thermo Scientific). All values were normalized based on protein concentrations.

### 2.5. Mitochondrial Membrane Potential Assay

Mitochondrial ATP production depends on mitochondrial activity, which can be assessed by measuring mitochondrial membrane potential using membrane potential-sensitive probes. In our experiments, following heat treatments CHO cells were labeled with 5,6-dichloro-2-[3-(5,6-dichloro-1,3-diethyl-1,3-dihydro-2H-benzimidazol-2-ylidene)-1-propen-1-yl]-1,3-diethyl-1 h-benzimidazolium, monoiodide (JC-1) dye (5 µM, 30 min, 37 °C in dark) according to the method used by Chinopoulos et al. with minor changes [29]. Stock solution was prepared in DMSO and diluted in culture medium. After labeling, cells were washed twice and resuspended in PBS. Then, fluorescence intensity was measured at 23 °C by PTI QuantaMaster spectrofluorometer (Horiba, Kyoto, Japan), with 488 nm excitation wavelength (5 nm slits); 535 (green) and 595 (red) emission wavelength (5 nm slits). Ratios of green and red fluorescence were determined and compared to control samples. 

### 2.6. Lipid Peroxidation Assay

Diphenyl-1-pyrenylphosphine (DPPP) obtained from Dojindo is a fluorescent probe capable of detecting peroxidation of cell membrane lipids. The probe is non-fluorescent but becomes fluorescent when oxidized. Assay was performed with minor changes based on a study by Wang et al. [30]. Stock solution (5 mM) was prepared in DMSO (Sigma-Aldrich) and stored at −20 °C. After heat treatments, sample plates were filled with 10 µM DPPP in medium and were incubated for 10 min at 37 °C. Then, plates were washed twice with PBS, and DPPP-labeled cells were collected after trypsinization. Fluorescence spectroscopy (352 nm excitation, 380 nm emission at 2 nm slit for both) using a PTI QuantaMaster spectrofluorometer (Horiba) was performed on resuspended cells. Oxidized DPPP fluorescence was measured before and after 95 mM BA treatment.

### 2.7. RNA Preparation, Sequencing and Bioinformatic Analysis

For the analysis of heat-induced transcriptome, cells were collected and prepared with a NucleoSpin RNA isolation kit for total RNA (Macherey-Nagel, Allentown, PA, USA) according to the manufacturer’s instructions. Briefly, 10^6^ cells were seeded on culture plates for 24 h prior to heat treatments which included 4 biological repeats of 4 sample groups (37 °C, 40 °C for 1 h, 6 h and 42.5 °C for 1 h). Then, cells were collected on an RNA binding column, which was followed by a DNase treatment to remove DNAs from samples. The quality and the quantity of the extracted RNAs were determined by a NanoDrop 2000/2000c Spectrophotometer and Qubit Fluorometer (both from Thermo-Fisher Scientific, Waltham, MA, USA), and by TapeStation (Agilent, Santa Clara, CA, USA). Sequencing library preparation was performed using a Tru Seq RNA Library Preparation Kit (Illumina, San Diego, CA, USA), according to the manufacturer’s instructions, and sequenced on an Illumina NextSeq instrument using 2 × 150 bp paired end sequencing chemistry. The bioinformatics analysis was carried out by the QC for Sequencing Reads, Trim Reads, RNASeq Analysis and the Differential Expression of RNA-seq modules of CLCBio Genomic Workbench (Qiagen, Hilden, Germany). Sequenced reads were mapped to Cricetulus griseus CHOK1GS_HDv1 genome version. The differential expression of genes was considered as significant when fold change > 1.5× and false discovery rate (FDR) < 0.05 selection criteria were fulfilled. The downstream gene overrepresentation analysis was done by Ingenuity Pathway Analysis software (IPA, Qiagen) using the mouse orthologs of the significant Chinese hamster genes. PCA and hierarchical cluster analysis of transcriptomic datasets were performed using MetaboAnalyst. In addition, out of the 16 measurement points only one control sample was excluded from the analysis due to its strong inconsistency with its group.

### 2.8. Lipidomics

CHO cells were left untreated or heat-treated at 40 °C and 42.5 °C for the specified time intervals, washed three times with cold PBS, collected in Eppendorf tubes (10^6^ cells per tube) and centrifuged. The pellets were shaken in 1 mL of methanol containing 0.001% butylated hydroxytoluene as an antioxidant, for 10 min, and centrifuged at 10,000× *g* for 5 min. The supernatant was transferred into a new Eppendorf tube and stored at −20 °C. Lipid standards were obtained from Avanti Polar Lipids (Alabaster, LA, USA). The solvents used for extraction and for MS analyses were of liquid chromatographic grade (Merck, Darmstadt, Germany) and Optima LCMS grade from Thermo Scientific.

Electrospray ionization mass spectrometry analyses were performed on an Orbitrap Fusion Lumos instrument (Thermo Fisher Scientific) equipped with a TriVersa NanoMate robotic nanoflow ion source (Advion BioSciences, Ithaca, NY, USA), as detailed in [31,32]. For MS measurements, 15 µL of lipid extract was diluted with 135 µL infusion solvent mixture (chloroform:methanol:iso-propanol 1:2:1, by vol.) containing an internal standard mix (Appendix A). Lipids were identified by the LipidXplorer software [33]. Data files generated by LipidXplorer queries were further processed by in-house Excel macros.

Lipid classes and species were annotated according to the classification systems for lipids [34]. In sum formulas for glycerolipids, e.g., PC(34:1), the total numbers of carbons followed by double bonds (db) for all chains are indicated. For sphingolipids, the sum formula, e.g., Cer(34:1:2), specifies first the total number of carbons in the long-chain base and the fatty acid moiety, then the sum of db in the long-chain base and the fatty acid moiety, followed by the sum of hydroxyl groups in the long-chain base and the fatty acid moiety.

### 2.9. Statistical Analyses

Prism 9.0 (Graphpad Software, San Diego, CA, USA) was used for statistical analyses of protein levels (*n* ≥ 5), survival (*n* ≥ 3), mitochondrial membrane potential (*n* ≥ 9) and lipid peroxidation (*n* ≥ 9) where results are given as means ± SD and *p*-value lower than 0.05 was considered statistically significant. Student’s *t*-tests were performed for comparisons of data. In the case of lipidomic experiments, data are presented as mean ± SEM. Student’s *t*-tests were performed for pairwise comparisons; significance was determined according to Storey and Tibshirani [35] and was accepted for *p* < 0.025 corresponding to a false discovery rate < 0.05. Multivariate statistical analysis of lipidomic and RNAseq datasets was performed using MetaboAnalyst [36]. 

## 3. Results

### 3.1. Duration of Fever-like Hyperthermia Determines the Threshold of Hsp Induction

Dose of stress (length and intensity) influences the cells’ response including Hsp synthesis. By using immunoblotting, we investigated how 40 °C, as fever-like heat exposure, affects Hsp induction in CHO cells. Based on previous results [4], two main chaperone proteins, Hsp25 and Hsp70, proved to be important candidates for assessing HS. Our results show that upon 40 °C heat treatment, Hsp25 induction was detected after 4 h of exposure (Figure 1a). In the case of Hsp70, no induction was seen even up to 12 h of exposure with or without recovery at 37 °C (Appendix A). Exposure to 40 °C with different time periods followed by 6 h recovery time was also investigated (Figure 1b). It is noted that up to 2 h Hsp25 induction could not be observed even after 6 h recovery at 37 °C.

For further experiments, we selected 1 h and 6 h of 40 °C heat treatments because of their difference in Hsp25 inducibility (Figure 1c,d). The effect of 1 h 40 °C HS is negligible while 6 h at 40 °C results in a significant induction of Hsp25 regardless of recovery time. A moderate 1 h HS at 42.5 °C followed by 6 h recovery at 37 °C resulted in a high level of Hsp25 induction (Figure 1b,d). 

### 3.2. Mild Heat-Induced Acquired Stress Tolerance Does Not Require Hsp Induction

It is known that cells might develop thermotolerance after a recovery period following a mild HS [4]. We investigated the kinetics of this process during a prolonged heat pretreatment followed by a challenging stress of either high dosage of heat, the membrane fluidizing agent BA or the oxidative agent *tert*-butyl hydroperoxide (TBHP).

We performed a colony formation assay in which CHO cells were primed with mild (40 °C for 1 h and 6 h) and moderate (42.5 °C for 1 h +/− recovery at 37 °C for 6 h) heat treatments followed by challenging stress including 46 °C for 20 min, 95 mM BA for 20 min or 250 µM TBHP for 3 h. A dosage of heat, BA and TBHP was selected at which cell survival of non-treated cells falls below 10% (Ref. [4] and Appendix A). Figure 2 shows cell survival after challenging stress as fold change compared to non-pretreated cells. For heat challenge, cellular survival correlated with the Hsp25 induction for cells treated at 40 °C for 6 h, but 42.5 °C for 1 h of treatment resulted in cell survival only after 6 h of recovery (Figure 2a). Cells acquired resistance against a membrane (by high dosage of BA) and oxidative (TBHP) challenge much more rapidly. A significant protection was observed after 1 h of priming at 40 °C, suggesting that de novo Hsp synthesis was not required in these cases. Interestingly, neither the length nor the temperature of the priming affected the level of cross resistance against BA and TBHP (Figure 2b,c).

BA increases membrane fluidity, thus altering the physical properties of membranes of cellular organelles. Mitochondria are some of the most sensitive organelles to membrane-perturbing agents [37] and it is known that mitochondrial oxidative phosphorylation is sensitive to local anesthetics such as BA [38], therefore we studied mitochondrial function.

As mitochondria are thought to be the powerhouse of the cell, it is logical to characterize the energetical state of the cell, that is, the level of cellular ATP. Luciferase-based assay showed no considerable change in cellular ATP levels at 40 °C and 42.5 °C heat treatment for 1 h, while there is only a slight but significant decrease at 40 °C for 6 h (Appendix A). 

We also intended to measure the mitochondrial membrane potential (ΔΨ_m_) to determine the effects of heat priming and BA challenge on mitochondria. For this purpose, we stained the cells with a fluorescent dye JC-1, the emission spectra of which change if mitochondria shift from an active to a more inactive state or vice versa. Our results indicate that heat treatment itself may alter the ΔΨ_m_ (Figure 3a), as already described before [28,39]. While BA challenge decreased ΔΨ_m_ significantly in all samples, protection by heat pretreatment could only be detected for cells primed with moderate heat (42.5 °C for 1 h).

Since HS and membrane fluidization induce the production of reactive oxygen species [40,41,42,43,44,45,46], we investigated how the priming temperature and BA challenge alter the amount of oxidized lipids. CHO cells were labeled with the lipid peroxidation-sensitive fluorescent dye diphenyl-1-pyrenylphosphine (DPPP) after priming. Fold change in fluorescence of oxidized DPPP was calculated by obtaining fluorescence before and after priming and after BA challenge (Figure 3b). Compared to non-primed cells, priming conferred a significant protection (48–65%) against lipid peroxidation elicited by BA challenge in all cases.

### 3.3. RNA Sequencing Data Reveal Distinct, Dose-Dependent Stress Transcriptomes 

To gain insight into the mechanism of the development of acquired stress tolerance, we conducted a transcriptomic analysis to obtain a global view of the effects of heat treatment. One prompt step of stress response is the altered gene expression, which normally contributes to the upregulated production of Hsps and other stress-related proteins and the hindering of less necessary protein synthesis [47,48,49,50].

We investigated the transcriptome of CHO cells following heat treatments (40 °C for 1 h, 6 h and 42.5 °C for 1 h) by RNA sequencing to identify molecular pathways associated with the different stress doses and acquired stress resistance.

We have individually compared the RNAseq transcriptome profiles of the three distinct heat shock-treated samples to the 37 °C controls. Out of the more than 20,000 genes of the Chinese hamster genome, we have altogether found 920 differentially expressed genes (DEGs) in the three treated–control sample pairs, which exhibited statistically significant change (FDR < 0.05) with at least a 1.5-fold gene expression difference (Appendix A). Table 1 represents a selection of those genes which changed most significantly upon the different heat treatments.

Based on principal component analysis (PCA) of the DEGs, the samples separated into four non-overlapping clusters, which corresponded to the different heat treatments, supporting the existence of unique gene expression patterns in all heat shock conditions (Figure 4a). 

The heatmap representation of hierarchical cluster analysis reflected the DEG patterns in all experimental conditions (Figure 4b). A mild HS exhibits a considerably distinct transcription profile compared to moderate heat exposure (42.5 °C, 1 h), although some of the DEGs exhibit a similar gene expression pattern.

The Venn diagram shows that with increasing intensity and/or length of heat treatment, the number of DEGs increases (Figure 4c). Under 40 °C mild stress, 4% and 32% of all DEGs changed significantly in 1 and 6 h, respectively, while under moderate heat treatment (42.5 °C, 1 h) ca. 75% of the DEGs were altered.

The top 50 most enriched pathways that involve these DEGs were identified by Ingenuity Pathway Analysis (IPA) (Appendix A). In the case of 40 °C, 1 h and 6 h heat treatments, genes of the most enriched pathways include many Hsps and other proteins related to HSR (e.g., aldosterone signaling in epithelial cells, unfolded protein response, protein ubiquitination pathway). IPA showed that 42.5 °C, 1 h treatment yielded higher enrichment for inflammation-related signaling pathways besides many which involve DEGs of Hsps (Figure 5 and Appendix A). Based on IPA prediction, one of the most significantly enriched pathways showing overall activation tendencies was the unfolded protein response, especially in cells heated at 40 °C for 6 h (Appendix A).

### 3.4. Lipidomic Data Reveal Distinct, Dose-Dependent Stress Lipidomes

To identify HS-related lipid signatures, we prepared total lipid extracts from untreated and heat-treated (40 °C for 1 and 6 h and 42.5 °C for 1 h) CHO cells, and performed high-sensitivity, high-resolution mass spectrometry-based shotgun lipidomic analysis. Approximately 380 lipid species were identified and quantified (Appendix A). Lipid compositional data (expressed as mol% of membrane lipids) were subjected to partial least squares discriminant analysis. The goodness-of-fit (R2) value of 0.98 and the goodness-of-prediction (Q2) value of 0.90 indicated that it was a reliable model to detect differences among the four sample groups and confirmed an adequate robustness. The well-distinguishable clusters (Figure 6a) demonstrated characteristic, dose-dependent changes in the lipidomes of CHO cells in response to HS.

Subsequently, we compared the molecular species patterns for all treatment groups; more than 200 lipid molecules changed significantly at least in one of the applied stress conditions relative to the unstressed control. The Venn diagram in Figure 6b shows that the longer mild HS (40 °C, 6 h) accounted for 62% of all significant alterations, whereas this number was 7% and 31% for the shorter mild (40 °C, 1 h) and moderate HS (42.5 °C, 1 h), respectively.

Further data analysis revealed several alterations with explanatory relevance. Table 2 represents a selection of those lipid species which changed significantly and contributed mostly to the global membrane lipidomic alterations, whereas the complete dataset is listed in Appendix A. The heat stress at 40 °C for 6 h caused comprehensive changes throughout several lipid classes, which reflected an adaptive response to the prolonged stress duration. The most remarkable feature was the significant increase in the relative concentration of disaturated and monounsaturated membrane lipid species (db ≤ 1), such as PC(32:0) or (PC(34:1) (Figure 7a and Appendix A), which was paralleled with a significant loss in polyunsaturated components (db ≥ 4), in particular, the arachidonic acid (20:4 n-6)-containing PC(36:4, 16:0/20:4) and PI(38:4, 18:0/20:4) species (Figure 7b and Appendix A). In addition, we documented a significant increase in the relative amount of lysoglycerophospholipid species, especially in LPC(16:0) and LPI(18:0) (Figure 7c and Appendix A). The moderate heat treatment (42.5 °C, 1 h) shared numerous changes with the prolonged mild stress (32% of all significant alterations), such as the relative increase in PC(32:0) and the decrease in PC(36:4) (Figure 7a,b). However, in contrast to the prolonged mild stress, the overall loss in polyunsaturated (PUFA)-containing components was significant only for species with db = 6 and db = 7, i.e., for those that contained docosahexaenoic acid (22:6 n-3) (Appendix A). Moreover, whereas the accumulation in the long-chain 24:1 fatty acid-containing ceramide species Cer(42:2:2) was sizeable in each conditions, the elevation in the major, palmitic acid-containing sphingolipid species Cer(34:1:2), hexosylceramide HexCer(34:1:2) and sphingomyelin SM(34:1:2) was specific for the 1 h heat stress at 42.5 °C (Figure 7d and Appendix A). Similarly to transcriptomic results, the 1 h HS at 40 °C induced only a subtle response at the level of the lipidome, mostly shared with the other treatments, and resulted, e.g., in the elevation of Cer(42:2:2) and LPI(18:0) (Figure 7c,d).

## 4. Discussion

In a multidisciplinary approach combining ultrasensitive fluorescence microscopy and lipidomics, we have previously revealed the molecular details of novel cellular “eustress”, when cells adapt to mild acute heat treatment by maintaining membrane homeostasis, activating lipid remodeling and redistributing chaperone proteins [4]. Previously, we characterized a short (20 min) incubation at 40 °C as mild heat dosage, which induced no Hsp synthesis, yet resulted in significant acquired thermotolerance [4]. For better understanding of febrile hyperthermia, in the current study we explored HSR in CHO cells during mild heat shock for 0–12 h. An increase in Hsp25 level was detected only after 2 h and 4 h of heat treatment, with or without recovery time, respectively. Nevertheless, even a longer heat treatment (≥6 h) yielded no detectable Hsp70 induction, indicating that the cells do not experience higher stress alertness [4]. Earlier, it was suggested that the accumulation of Hsp chaperones is a prerequisite to establish effective acquired thermotolerance [51]. A recent high-throughput quantitative proteomics and targeted mRNA quantification in Jurkat T lymphocytes revealed that a moderate heat treatment of 4 h, 41 °C causes only a minor across-the-board mass loss in housekeeping proteins which is matched by a mass gain in a few Hsps, predominantly cytosolic [52]. Our Western blot experiments, however, suggest that during the initial few hours of a milder heat treatment the perturbation in the protein production is negligible regardless of the preexisting mRNAs. Previous investigations on avian (chicken reticulocytes) and mammalian (Chinese hamster fibroblast) cells also indicated that a mild temperature treatment up to 41 °C does not necessarily affect protein production [53,54].

A longer mild (40 °C for 6 h) or a short moderate (42.5 °C for 1 h) heat pretreatment before subsequent prompt, lethal, protein-denaturing heat shock produced acquired stress tolerance only if the corresponding Hsp25 level was higher, suggesting the requirement of the classical Hsp response. Indeed, it was previously reported that heat preconditioning at a mild temperature (40 °C for 3 h) in mammalian cells led to the development of thermotolerance, which was associated with an increase in the expression of several Hsps including Hsp72 [55,56]. The expression of Hsps 27, 32, 72 and 90 was significantly increased after 2 or 3 h at 40 °C in HeLa and CHO cells in agreement with our results of Hsp25 induction (Figure 1). Our current and previous studies in mammalian [4] and bacterial [57] cells and others’ observations in plants [58], however, show that membrane protective stress tolerance develops in the absence of the induction of stress proteins which indeed are required for a more severe, protein-denaturing heat treatment. These previous findings have already highlighted the importance of membrane retailoring under such conditions.

In order to test how mild hyperthermia could protect membrane homeostasis, we used benzyl alcohol, a membrane fluidizer agent that decreases the viscosity of lipid bilayers, thus mimicking the effect of higher temperatures without unfolding proteins [28]. Interestingly, a preconditioning with either mild or moderate heat treatment resulted in acquisition of BA tolerance independently of Hsp25 induction (Figure 2). BA can also spatially compete with lipids that normally surround intrinsic proteins, which might result in deterioration of their function [26,27]. Accordingly, it has an adverse effect on the members of the mitochondrial electron transport chain [38]. For example, Armston et al. found that BA-induced membrane fluidization coincidentally changes with the arrest of mitochondrial succinate oxidation [59]. The latter leads to succinate accumulation [60], which triggers the production of reactive oxygen species (ROS) [61]. Another study reported that BA damages mitochondria through ROS induction, and as a result of mitochondrial impairment, further ROS production takes place and creates a positive feedback loop in the direction of cellular death [62]. Mitochondrial lipids, especially cardiolipin and PUFAs, are prone to lipid peroxidation caused by ROS [63]. Additionally, Sergent et al. linked the effect of ethyl alcohol to lipid peroxidation through ROS production [45]. The amount of lipid peroxidation of primed cells following BA challenge is consistent with the cellular survival which can be regarded as another aspect of acquired stress tolerance (Figure 3b vs. Figure 2). Since mild heat priming (in this study) does not protect cells against severe heat-induced protein denaturation but protect against BA challenge, we can probably rule out BA-induced protein denaturation. As mitochondria are the main source of cellular ATP, it is noteworthy that no significant change in the ATP level could be detected between cells incubated at 37 °C and 40 °C (Appendix A). We also found that mitochondrial membrane potential was increased by a short dosage of mild heat (Figure 3a) which was previously observed only for higher temperatures (42–45 °C) [28,55]. Mitochondrial hyperpolarization represents an early and reversible step in apoptosis, peaking at the dose of the stressors that elicit maximum Hsp response and it could serve as a key event in the stress signaling of K562 cells [28]. As in a study by others [62], our results indicate a decrease in the ΔΨm after high dosage of BA challenge. Interestingly, in a previous study we found an opposite effect of BA and another fluidizing agent heptanol but in that case these agents were used in a much lower concentration [28].

Mild hyperthermia also protects against oxidative stress (Figure 2c). Interestingly, even the lowest dosage of mild hyperthermic pretreatment resulted in a significant protection against *tert*-butyl hydroperoxide (TBHP) treatment which generates a broad spectrum of free radicals in different intracellular compartments. As cytoprotective agents, Hsps enable cells to resist as well as recover from oxidative stress at many levels [64]. Since lipid peroxidation is considered to be one of the primary reasons for cell damage during oxidative stress, the translocation of preexisting Hsps to membranes [20,65] could be one possible explanation, especially since Hsp relocalization was also observed in our previous study in CHO cells [4]. However, the loss in polyunsaturated species during mild hyperthermia (see below) could also significantly stabilize membranes against oxidative damage.

Upon fever-like conditions, the expression of a large number of genes is affected together with many HSR- or UPR-related RNAs [25,66,67]. Ingenuity Pathway Analysis (IPA) of our RNAseq data revealed several enriched pathways in a mild heat shock condition, out of which many are related to stress response and contain HSR- and UPR-associated DEGs (Appendix A). A moderate heat condition, in addition, involved the enrichment of inflammation-associated pathways (Appendix A). In agreement with a previous study [68], in our transcriptomics experiments UPR was the most significantly influenced pathway under a longer mild hyperthermia condition. In the case of 40 °C, 6 h heat treatment, the IPA MAP tool predicted a substantial activation of the entire UPR pathway due to the upregulation of the key regulator HspA5 (alias BIP). Nevertheless, this prediction cannot be seen for the 40 °C, 1 h condition, possibly because of the too short treatment, while for 42.5 °C, 1 h, other cellular processes might be dominant. IPA simulation suggests that mild heat shock induces all the three branches of UPR, which presumes the importance of endoplasmic reticulum stress in acquiring stress tolerance (Appendix A).

The ER stress pathway has already been reported to be induced partially or fully as a consequence of mild hyperthermia in parallel with induction of Hsp70 in AD293 cells [68]. They postulated that the activation of the ER stress pathway in parallel with the HSR orchestrates adaptation to febrile hyperthermia that occurs because of disease and infection. In another study, it was shown that preexposure to mild hyperthermia (40 °C for 3 h) alleviates the induction of cytotoxicity and ER stress by severe hyperthermia (42–43 °C) and protects HeLa cells against ER stress-induced apoptosis [67]. This protective effect could be abrogated by shRNA-mediated depletion of Hsp72. Our results suggest that in CHO cells the induction of ER stress response is required to develop tolerance to membrane-perturbing stress, but it is not sufficient to protect cells against a more severe heat shock, which requires Hsp induction (Figure 2). 

Protein quality control and lipid metabolism are intimately connected at both cellular and molecular levels. Besides the accumulation of unfolded proteins in the lumen of the ER, modulations in the physicochemical properties of the ER membrane, due to lipid imbalances, serve as activating signals for the UPR [69]. Previously, we showed that HS treatment gave rise to distinct CHO lipidomic fingerprints, in a temperature-dependent manner, after a very short, 20 min heat exposure [4]. We postulated that the structural changes in cellular membranes are linked to HS signaling by specific sensors or sensor networks. Indeed, ER-resident proteins were identified that sense bulk membrane lipid properties to maintain lipid homeostasis [14]. However, only sensors that detect fluidity decrease have been identified to date. We propose that either a new set of ER sensors exists for detecting fluidity increases or the currently identified sensors are also capable of sensing an upshift in fluidity. IRE1 is a unique protein in this respect because it senses misfolded proteins through its ER luminal domain, but senses lipid bilayer stress by its transmembrane domain as well. 

Here, we investigated different layers of stress response elicited by different HS dosages. Importantly, the number of significant differences for the prolonged mild heat and the moderate HS showed a reversed distribution pattern for transcriptomic vs. lipidomic data, i.e., the milder but prolonged stress left more time to execute adaptive membrane lipid changes, whereas the acute but harsher stress provoked the expression of genes for a broader arsenal of protective pathways. 

To maintain membrane fluidity, fatty acid saturation is known to be tightly regulated to ensure cell functionality and vitality [11]. The significant elevation in the relative concentration of saturated membrane lipid species together with a significant loss in polyunsaturated species represented an overall membrane rigidification in response to the fluidizing HS. To execute such changes, concerted actions of enzymes of de novo lipid synthesis, subsequent deacylation/reacylation cycles and/or phospholipase actions are required. The loss in PUFA-containing lipids, in particular those that contain arachidonic acid, could be the result of phospholipase A2 induction. The enzyme selectively removes the PUFA from the sn2 position of mammalian phospholipids and generates the corresponding lysolipid; this is in agreement with the increase in lysolipids, e.g., LPC(16:0) (from PC(36:4)) or LPI(18:0) (from PI(38:4)). The loss in PI(38:4) can be the result of phospholipase C activity, too. The activations of phospholipase A2 and C were previously demonstrated in B16 cells subjected to different membrane-perturbing stresses [70]. Alternatively, downregulation of specific lysolipid acyltransferases, which incorporate PUFAs into phospholipids during membrane maturation, could also contribute to the relative accumulation of lysolipids and lowering of polyunsaturated species. Indeed, our lipidomic and transcriptomic observations are in agreement with a previous finding which showed that a decrease in membrane phospholipid unsaturation, provoked by stearoyl-CoA desaturase 1 knockdown, induced UPR in HeLa cells, and this could be synergistically enhanced by LPCAT3 knockdown [71]. Importantly, the products of phospholipase A2 action, free PUFAs and lysolipids themselves possess various signaling properties. In the present study, we found that LPI(18:0) displayed a significant increase, especially upon the prolonged mild HS. The discovery that the orphan G protein-coupled receptor GPR55 acts as the specific receptor for LPI has fueled novel interest in this lysolipid [72]. The LPI/GPR55 axis was suggested to play an important role in different physiological and pathological contexts. A recent study provided a novel protective effect of GPR55 agonists on ER stress-induced apoptosis in β-cells [73]. Therefore, we propose that the HS-induced LPI elevation could participate in CHO cell UPR signaling. On the other hand, due to their positive curvature-inducing property, the relative accumulation of lysophospholipids generates lipid bilayer stress, and might ultimately lead to UPR activation. Recently, it was also reported that macrophage IRE1 plays an unprecedented role in regulating phosphatidylinositide-derived signaling lipid metabolites (PIPs) and has a profound impact on the downstream mTOR-AKT signaling to control cellular growth [74]. PI(38:4) is the major source of phosphoinositides, therefore, its heat-induced decrease might modulate the ratio of different PIPs, thereby influencing the PI3K/mTOR/AKT signaling axis [19]. Such crosstalk between UPR and mTOR pathways could conceivably contribute to the effective cellular protection in HSR.

The changes that occurred in response to the moderate heat stress, especially the relative increase in different sphingolipids, suggest prompt membrane microdomain reorganization, as was previously shown for different cell types subjected to harsher temperature stress [70,75].

The modulation of the membrane structure by heat has been observed in many organisms before [76]. For instance, acclimation to different temperature environments causes adaptive changes in the lipid composition of cell membranes of fishes [77]. These changes in brain lipids have been associated with the maintenance of neuronal biophysical membrane properties at different temperatures, with the aim of keeping an appropriate cell signaling and correct physiology [78]. This phenomenon, called “homeoviscous adaptation” [79] of biomembranes, is a direct consequence of the classical fluid mosaic model [80] and its updated “dynamically structured” version [81]. Membranes need to be maintained in a functionally competent fluid state. Any perturbation of this structure should be quickly repaired to maintain important membrane-associated biological processes. Alternatively, to facilitate rapid retailoring of lipid composition, preexisting Hsps could temporarily associate with membranes and can reestablish the fluidity and bilayer stability and thereby restore the membrane functionality during stress conditions [82].

The maintenance of membrane homeostasis during diseases is especially important since under these circumstances cells may not be able to respond to stress conditions. Therefore, the modulation of membrane structure and dynamics could be an important therapeutic target. Such a “membrane-lipid therapy” approach was suggested to have potential for the treatment of metabolic, neurodegenerative, cancer and many other diseases [83]. Small molecules have already been suggested to therapeutically modulate the bilayer structure to treat type 2 diabetic complications, cancer, neurodegenerative diseases, etc. [84,85]. Mild hyperthermia, however, could be a viable alternative to such approaches [86]. The present study sheds light on the cellular and molecular mechanism of action for small molecules described before and thermal eustress. In this context, the use of electromagnetic fields to increase temperature in tumors and increase the efficacy of chemotherapy has been used for several years without clear elucidation of its mechanism of action [87]. The present study provides rationale for these therapeutic approaches, connecting thermal regulation with membrane lipid composition modulation and Hsp expression changes that control protein stability and cell signaling.

## 5. Conclusions

Stress response pathways have evolved to maintain cellular homeostasis and to ensure the survival of organisms under changing environmental conditions. Whereas severe stress is detrimental, mild stress can be beneficial for health and survival. In this study, we aimed to shed more light onto the cellular stress response, particularly in the case of fever-type mild heat, as distinct cellular tools sequentially engage in the coping mechanism. Besides understanding the fundamental basis of stress responses, novel therapeutic applications might arise from studying fever-type HS. For this reason, we used CHO cells to delineate the transition of cellular tools in response to prolonged stress by lipidomics and RNAseq experiments along with Hsp induction, survival and lipid peroxidation measurements.

Although the universally conserved heat shock response regulated by transcription factor HSF-1 has been implicated as an effector mechanism, the role and possible interplay with other cellular processes, such as UPR, remain poorly understood. 

Our study reveals that prolonged mild heat can similarly initiate ER-related signaling cascades resulting in lipid rearrangements and ultimately in an elevated resistance against membrane stress. Acquired stress resistance is induced in CHO cells following a mild hyperthermia which also results in a profound membrane lipid retailoring in a stress dosage-dependent manner. Although the complex chaperome network organization may contribute to this phenomenon [88], our findings demonstrate that UPR and membrane homeostatic changes are required for acquired membrane-perturbing stress resistance following a fever-type non-protein denaturing preconditioning, while Hsps are vitally important to survive a more severe, protein-denaturing heat shock. To generalize our results for different cells and tissues, broader systematic studies are needed in the future.

## Figures and Tables

**Figure 1 biomedicines-10-01172-f001:**
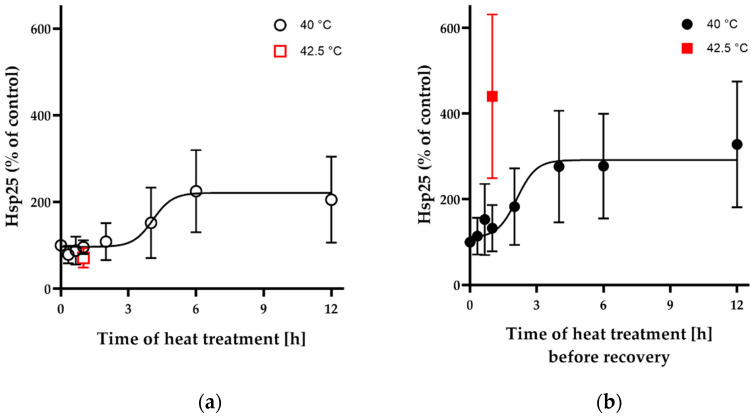
The effect of heat dosage on Hsp induction measured by Western blotting. (**a**,**b**) Time-dependent induction of Hsp25 upon mild heat treatment of CHO cells. Samples (40 °C and 42.5 °C denoted by black circles and red squares, respectively) were collected (**a**) right after (empty marks) HS or (**b**) following a 6 h recovery at 37 °C (solid marks). (**c**) Blot images of Hsp25, Hsp70 and GAPDH protein levels following a heat priming at 40 °C for 1 h or 6 h and 42.5 °C for 1 h without or with 6 h recovery (R) at 37 °C. (**d**) Relative quantification for Hsp25 induction in bar chart, in which recovery is indicated by checkerboard pattern. All data represent the means ± SD; *n* ≥ 5, *p* < 0.05 was considered statistically significant. Paired *t*-test was used for statistical comparisons, in which *, ° and + denote significant difference compared to 37 °C; 40 °C, 1 h with recovery; or 40 °C, 6 h with recovery, respectively. See experimental setup and blot images in Appendix A.

**Figure 2 biomedicines-10-01172-f002:**
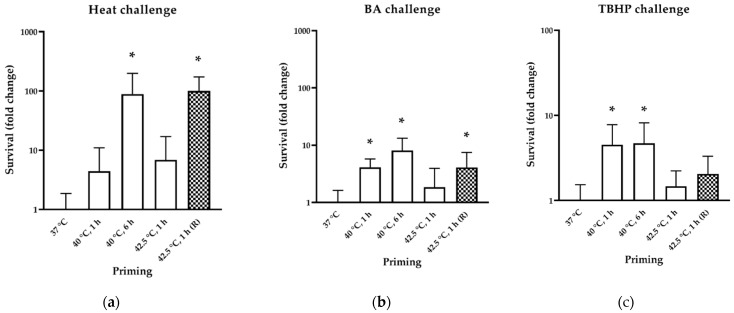
Acquired stress tolerance of heat-primed CHO cells. The quantitative analysis is based on colony formation assay. Primings were 1, 6 h at 40 °C or 42.5 °C for 1 h without or with 6 h recovery (R/checkerboard pattern) at 37 °C, which was followed by (**a**) a heat challenge with 46 °C for 20 min, (**b**) membrane fluidizing challenge by 95 mM BA for 20 min or (**c**) oxidative challenge by 250 µM TBHP for 3 h. Survival is represented as fold change compared to non-primed controls (37 °C). All data represent the means ± SD; *n* ≥ 3, *p* < 0.05 was considered statistically significant. Non-paired *t*-test was used for statistical comparisons, in which * denotes significant difference compared to 37 °C samples that received heat or BA challenge. See experimental setup and colony formation assay images in Appendix A.

**Figure 3 biomedicines-10-01172-f003:**
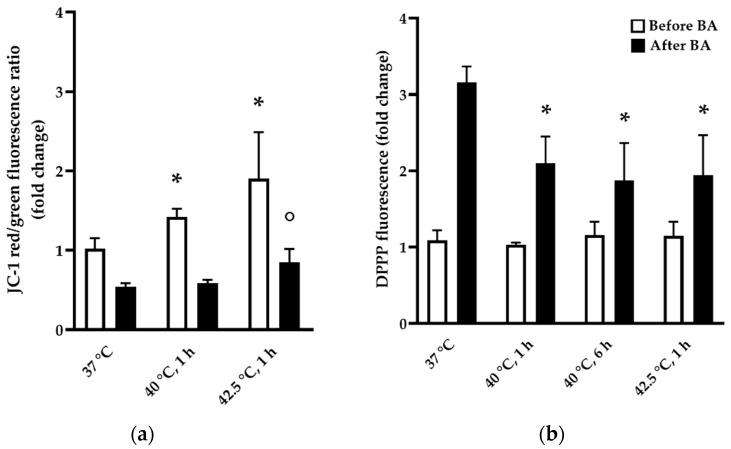
The effect of heat priming on mitochondrial membrane potential and lipid peroxidation of CHO cells. (**a**) Change in mitochondrial membrane potential measured by fluorescent JC-1 dye. Cells were primed at 40 °C for 1 h and 42.5 °C for 1 h, then fluorescence was read before (empty bars) and after (solid bars) addition of a challenging dose of 95 mM BA. Fold change was calculated in comparison to non-treated 37 °C. (**b**) CHO cells were primed at 40 °C for 1 h and 6 h and at 42.5 °C for 1 h, followed by lipid peroxidation-sensitive DPPP fluorescence measurement (empty bars). After heat pretreatments a challenging dose of 95 mM BA was introduced to the cells (solid bars). In (**a**,**b**), all data represent the means ± SD; *n* ≥ 9, *p* < 0.05 was considered statistically significant. Paired *t*-test was used for statistical comparisons. In (a), * and ° denote statistically significant difference compared to 37 °C and 37 °C + BA, respectively, while in (**b**), significant difference was compared to 37 °C + BA and denoted by *.

**Figure 4 biomedicines-10-01172-f004:**
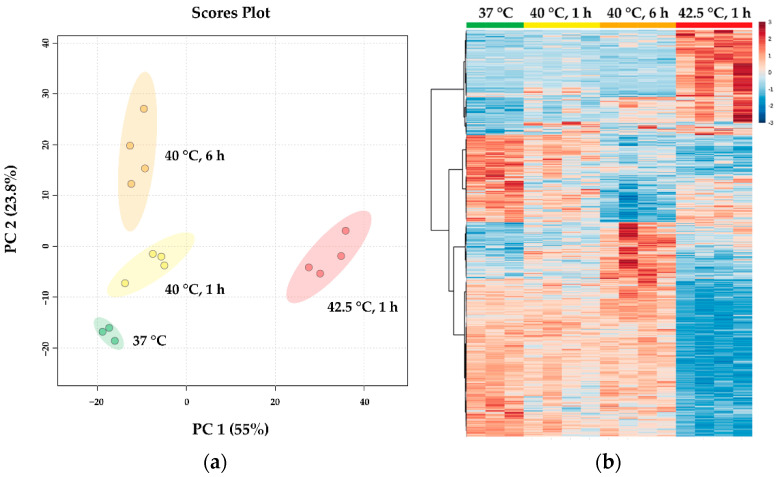
RNA sequencing data reveal distinct stress transcriptome upon different stress conditions. (**a**) RNAseq samples of CHO cells distinguished by principal component analysis. (**b**) Heatmap representation of hierarchical clustering. (**c**) Expression of total of 920 genes had significantly changed by different doses of mild (40 °C, 1 h and 6 h) and moderate (42.5 °C, 1 h) heat treatments in pairwise comparison to controls (37 °C)—shown in Venn diagram.

**Figure 5 biomedicines-10-01172-f005:**
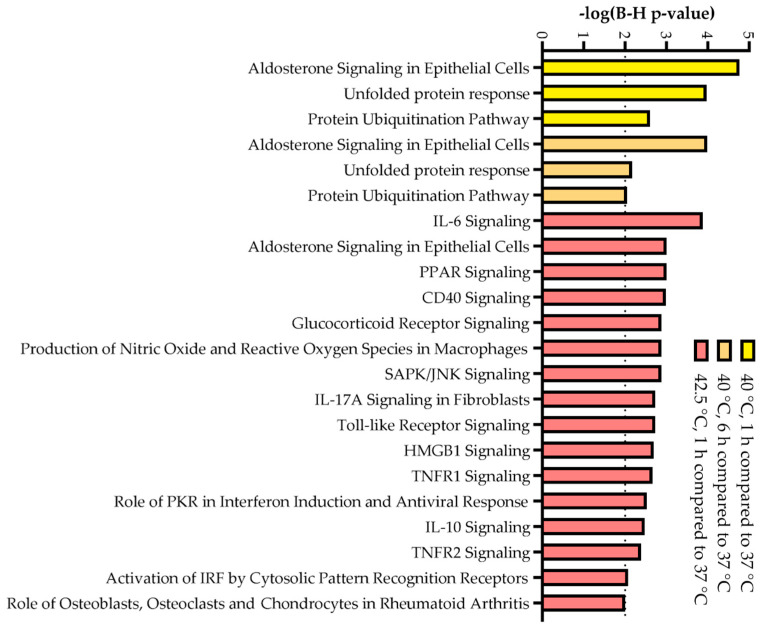
The most significantly enriched canonical pathways (−log(B-H *p*-value) > 2.0) in heat-treated CHO cells derived from Ingenuity Pathway Analysis (IPA).

**Figure 6 biomedicines-10-01172-f006:**
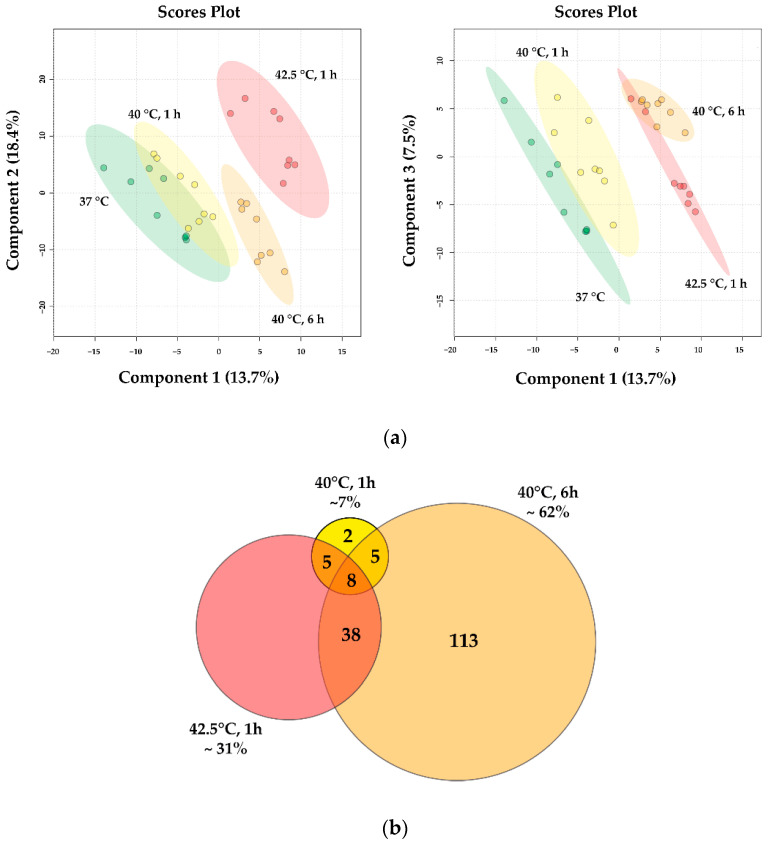
Heat-induced lipidomic changes. CHO cells were left untreated or subjected to 40 °C for 1 or 6 h or 42.5 °C for 1 h. (**a**) Partial least squares discriminant analysis score plots of lipidomic dataset based on relative concentration values. Circles display 95% confidence regions. The model was validated by a 2000-time permutation test (*p* = 0.0015). (**b**) Venn diagram displaying the number of statistically different components.

**Figure 7 biomedicines-10-01172-f007:**
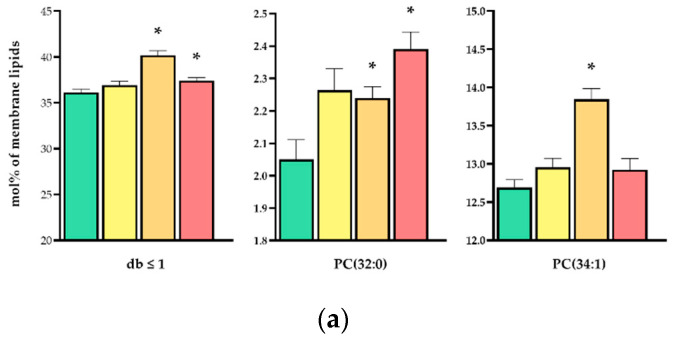
Heat-induced lipidomic changes. CHO cells were left untreated or subjected to 40 °C for 1 or 6 h or 42.5 °C for 1 h. (**a**–**d**) Changes in the relative concentration (**a**) of disaturated and monounsaturated membrane lipid species (db ≤ 1), (**b**) in polyunsaturated components (db ≥ 4), (**c**) in lysolipid species and (**d**) in sphingolipids. Data are expressed as mean ± SEM, *n* = 8; * denotes *p* < 0.025 compared to the untreated control. db, number of double bonds; PC, phosphatidylcholine; PI, phosphatidylinositol; LPC and LPI, the corresponding lyso species; Cer, ceramide; SM, sphingomyelin; HexCer, hexosylceramide.

**Table 1 biomedicines-10-01172-t001:** The mouse orthologs of the top 20 most increased (red) and decreased (green) genes exhibiting statistically significant expression change (fold change > 1.5, FDR < 0.05) *. The complete dataset with the original Chinese hamster genes is listed in Appendix A.

(**a**) The strongest expression change for sample pair 40 °C, 1 h compared to 37 °C.			
**Symbol** **(Mouse Orthologs)**	**Entrez Gene Name**	**Fold Change**	**FDR**
HSPA1A/HSPA1B	heat shock protein family A (Hsp70) member 1A	9.82	0.00 × 10^0^
GLYATL3	glycine-N-acyltransferase like 3	4.65	7.30 × 10^3^
ZFAND2A	zinc finger AN1-type containing 2A	3.29	0.00 × 10^0^
DNAJB1	DnaJ heat shock protein family (Hsp40) member B1	3.17	0.00 × 10^0^
IL11	interleukin 11	2.41	9.05 × 10^−9^
CLK1	CDC like kinase 1	2.35	0.00 × 10^0^
HSPB1	heat shock protein family B (small) member 1	2.35	9.02 × 10^−12^
HSPH1	heat shock protein family H (Hsp110) member 1	2.23	2.18 × 10^−8^
JUN	Jun proto-oncogene, AP-1 transcription factor subunit	2.09	0.00 × 10^0^
IER5L	immediate early response 5 like	2.08	3.18 × 10^−12^
RSRP1	arginine and serine rich protein 1	2.03	0.00 × 10^0^
RBAK	RB associated KRAB zinc finger	1.86	6.28 × 10^−7^
BAG3	BCL2 associated athanogene 3	1.84	4.00 × 10^−8^
DEDD2	death effector domain containing 2	1.83	9.10 × 10^−7^
CEP295NL	CEP295 N-terminal like	1.77	1.83 × 10^−4^
HSPA8	heat shock protein family A (Hsp70) member 8	1.65	6.05 × 10^−10^
TIPARP	TCDD inducible poly(ADP-ribose) polymerase	1.63	3.27 × 10^−9^
DUSP8	dual specificity phosphatase 8	1.61	1.10 × 10^−7^
NUAK2	NUAK family kinase 2	1.60	1.00 × 10^−2^
CCDC117	coiled-coil domain containing 117	1.58	1.73 × 10^−5^
RIN2	Ras and Rab interactor 2	−309.83	4.55 × 10^−9^
HOXB3	homeobox B3	−2.33	5.07 × 10^−3^
LCMT2	leucine carboxyl methyltransferase 2	−2.30	1.39 × 10^−4^
NAA20	N(alpha)-acetyltransferase 20, NatB catalytic subunit	−2.13	8.40 × 10^−3^
PLK2	polo like kinase 2	−1.90	0.00 × 10^0^
PTP4A1	protein tyrosine phosphatase 4A1	−1.73	4.00 × 10^−2^
MARCHF7	membrane associated ring-CH-type finger 7	−1.66	2.68 × 10^−9^
GAS8	growth arrest specific 8	−1.58	4.63 × 10^−3^
FDXACB1	ferredoxin-fold anticodon binding domain containing 1	−1.52	9.58 × 10^−3^
(**b**) The strongest expression change for sample pair 40 °C, 6 h compared to 37 °C.			
**Symbol** **(Mouse Orthologs)**	**Entrez Gene Name**	**Fold Change**	**FDR**
CRNKL1	crooked neck pre-mRNA splicing factor 1	187.36	2.00 × 10^−2^
CEL	carboxyl ester lipase	13.19	4.00 × 10^−2^
DNAH10	dynein axonemal heavy chain 10	8.81	2.00 × 10^−2^
GLYATL3	glycine-N-acyltransferase like 3	7.06	5.88 × 10^−6^
CCDC154	coiled-coil domain containing 154	7.04	1.04 × 10^−3^
WDR66	WD repeat domain 66	6.16	4.99 × 10^−6^
HSPA1A/HSPA1B	heat shock protein family A (Hsp70) member 1A	5.50	1.05 × 10^−13^
ARHGEF25	Rho guanine nucleotide exchange factor 25	4.85	2.00 × 10^−2^
PTP4A1	protein tyrosine phosphatase 4A1	4.81	5.93 × 10^−3^
COL24A1	collagen type XXIV alpha 1 chain	4.69	5.00 × 10^−2^
SPEF1	sperm flagellar 1	4.50	9.98 × 10^−3^
HHIPL2	HHIP like 2	4.19	3.53 × 10^−6^
TNFRSF25	TNF receptor superfamily member 25	3.77	1.00 × 10^−2^
MTMR7	myotubularin related protein 7	3.24	3.76 × 10^−4^
HSPB1	heat shock protein family B (small) member 1	3.05	0.00 × 10^0^
RGL3	ral guanine nucleotide dissociation stimulator like 3	3.04	1.00 × 10^−2^
RHBDL1	rhomboid like 1	2.94	1.00 × 10^−2^
HDAC9	histone deacetylase 9	2.92	8.08 × 10^−5^
TMED6	transmembrane p24 trafficking protein 6	2.89	3.00 × 10^−2^
ANKRD13D	ankyrin repeat domain 13D	2.87	3.01 × 10^−3^
RIN2	Ras and Rab interactor 2	−42.68	6.65 × 10^−6^
IYD	iodotyrosine deiodinase	−8.96	6.90 × 10^−3^
NAA20	N(alpha)-acetyltransferase 20, NatB catalytic subunit	−5.69	1.33 × 10^−12^
THEMIS2	thymocyte selection associated family member 2	−5.08	3.00 × 10^−2^
FNDC10	fibronectin type III domain containing 10	−4.07	8.76 × 10^−4^
TENM3	teneurin transmembrane protein 3	−3.66	4.01 × 10^−4^
SDCBP2	syndecan binding protein 2	−3.60	7.06 × 10^−3^
Ccdc74a	coiled-coil domain containing 74A	−3.14	4.00 × 10^−2^
CAPN8	calpain 8	−3.11	2.45 × 10^−5^
Dnajb3	DnaJ heat shock protein family (Hsp40) member B3	−2.98	1.98 × 10^−4^
Gm1110	predicted gene 1110	−2.98	2.35 × 10^−4^
Ccl2	chemokine (C-C motif) ligand 2	−2.97	0.00 × 10^0^
Ccl7	chemokine (C-C motif) ligand 7	−2.95	5.68 × 10^−11^
EGR1	early growth response 1	−2.88	0.00 × 10^0^
EGR3	early growth response 3	−2.82	0.00 × 10^0^
CST6	cystatin E/M	−2.78	4.00 × 10^−2^
CXCL10	C-X-C motif chemokine ligand 10	−2.74	3.00 × 10^−2^
CCR7	C-C motif chemokine receptor 7	−2.67	0.00 × 10^0^
HNRNPDL	heterogeneous nuclear ribonucleoprotein D like	−2.63	0.00 × 10^0^
BCLAF1	BCL2 associated transcription factor 1	−2.53	5.19 × 10^−3^
(**c**) The strongest expression change for sample pair 42.5 °C, 1 h compared to 37 °C.			
**Symbol** **(Mouse Orthologs)**	**Entrez Gene Name**	**Fold Change**	**FDR**
HSPA1A/HSPA1B	heat shock protein family A (Hsp70) member 1A	464.60	0.00 × 10^0^
CRNKL1	crooked neck pre-mRNA splicing factor 1	182.50	1.00 × 10^−2^
GAB2	GRB2 associated binding protein 2	93.26	4.00 × 10^−2^
KDF1	keratinocyte differentiation factor 1	58.54	4.73 × 10^−5^
AKAP5	A-kinase anchoring protein 5	52.76	1.14 × 10^−7^
ALDH8A1	aldehyde dehydrogenase 8 family member A1	41.51	2.77 × 10^−4^
FOS	Fos proto-oncogene, AP-1 transcription factor subunit	30.14	0.00 × 10^0^
LRIT3	leucine rich repeat, Ig-like and transmembrane domains 3	26.41	5.34 × 10^−13^
MAB21L1	mab-21 like 1	23.95	3.27 × 10^−3^
KLB	klotho beta	22.63	1.07 × 10^−6^
TRIM72	tripartite motif containing 72	22.12	4.56 × 10^−3^
AREG	amphiregulin	21.64	0.00 × 10^0^
HAS1	hyaluronan synthase 1	20.11	6.67 × 10^−3^
ZFAND2A	zinc finger AN1-type containing 2A	16.36	0.00 × 10^0^
CEL	carboxyl ester lipase	15.64	2.00 × 10^−2^
PDE8A	phosphodiesterase 8A	15.04	2.00 × 10^−2^
DNAJB1	DnaJ heat shock protein family (Hsp40) member B1	11.82	0.00 × 10^0^
LRRC4	leucine rich repeat containing 4	10.84	5.73 × 10^−3^
NLGN3	neuroligin 3	10.25	8.92 × 10^−4^
HTR2B	5-hydroxytryptamine receptor 2B	9.09	1.00 × 10^−2^
RIN2	Ras and Rab interactor 2	−29.93	7.82 × 10^−7^
CXCL2	C-X-C motif chemokine ligand 2	−8.46	0.00 × 10^0^
BCLAF1	BCL2 associated transcription factor 1	−7.79	2.97 × 10^−5^
DDAH2	dimethylarginine dimethylaminohydrolase 2	−5.82	2.00 × 10^−2^
TLR6	toll like receptor 6	−4.96	1.62 × 10^−6^
Gm11214	glyceraldehyde-3-phosphate dehydrogenase pseudogene	−4.70	3.00 × 10^−2^
PTP4A1	protein tyrosine phosphatase 4A1	−4.37	1.19 × 10^−14^
HPS6	HPS6 biogenesis of lysosomal organelles complex 2 subunit 3	−3.98	0.00 × 10^0^
DDX28	DEAD-box helicase 28	−3.94	0.00 × 10^0^
OSBPL7	oxysterol binding protein like 7	−3.85	0.00 × 10^0^
HOXD4	homeobox D4	−3.83	1.22 × 10^−5^
HOXA1	homeobox A1	−3.55	7.93 × 10^−3^
ABTB1	ankyrin repeat and BTB domain containing 1	−3.53	1.38 × 10^−12^
LFNG	LFNG O-fucosylpeptide 3-beta-N-acetylglucosaminyltransferase	−3.45	0.00 × 10^0^
NFKBIA	NFKB inhibitor alpha	−3.45	0.00 × 10^0^
CSF3	colony stimulating factor 3	−3.44	6.38 × 10^−3^
AMT	aminomethyltransferase	−3.39	4.60 × 10^−14^
Iigp1	interferon inducible GTPase 1	−3.38	6.35 × 10^−3^
MYORG	myogenesis regulating glycosidase (putative)	−3.37	6.90 × 10^−8^
EPHB3	EPH receptor B3	−3.32	2.74 × 10^−7^

* The overall number of significantly decreasing genes in the case of sample pair 40 °C, 1 h compared to 37 °C was only 9.

**Table 2 biomedicines-10-01172-t002:** Selected, significantly changing membrane lipid species.

Lipid Species	37 °C	40 °C, 1 h	40 °C, 6 h	42.5 °C, 1 h
Glycerophospholipids				
PC(32:1)	2.568 ± 0.023	2.564 ± 0.025	2.680 ± 0.016 *	2.537 ± 0.040
PC(32:0)	2.050 ± 0.061	2.264 ± 0.066	2.239 ± 0.036 *	2.391 ± 0.053 *
PC(34:2)	2.949 ± 0.029	2.936 ± 0.030	3.115 ± 0.025 *	2.842 ± 0.044
PC(34:1)	12.691 ± 0.102	12.957 ± 0.117	13.844 ± 0.144 *	12.924 ± 0.145
PC(36:4)	2.894 ± 0.060	2.847 ± 0.075	2.487 ± 0.031 *	2.664 ± 0.054 *
PC(36:1)	1.942 ± 0.028	1.970 ± 0.023	2.172 ± 0.051 *	1.892 ± 0.028
PC(38:6)	1.468 ± 0.017	1.460 ± 0.037	1.322 ± 0.035 *	1.256 ± 0.024 *
PC(38:5)	2.438 ± 0.046	2.434 ± 0.075	1.953 ± 0.050 *	2.231 ± 0.052 *
PC(38:4)	1.657 ± 0.042	1.685 ± 0.057	1.359 ± 0.042 *	1.588 ± 0.045
PC(40:6)	0.778 ± 0.009	0.770 ± 0.018	0.688 ± 0.022 *	0.651 ± 0.013 *
PC(O-32:0)	0.188 ± 0.007	0.202 ± 0.012	0.288 ± 0.013 *	0.212 ± 0.010
PC(O-34:1)	1.755 ± 0.062	1.819 ± 0.056	2.172 ± 0.058 *	1.941 ± 0.053
PC(O-36:1)	0.229 ± 0.016	0.236 ± 0.017	0.329 ± 0.016 *	0.231 ± 0.015
PC(O-38:5)	1.174 ± 0.023	1.219 ± 0.020	1.069 ± 0.026 *	1.298 ± 0.020 *
PE(34:1)	0.730 ± 0.011	0.728 ± 0.018	0.805 ± 0.019 *	0.671 ± 0.016 *
PE(38:6)	0.406 ± 0.014	0.380 ± 0.006	0.339 ± 0.006 *	0.355 ± 0.009 *
PE(38:5)	0.714 ± 0.028	0.659 ± 0.013	0.571 ± 0.013 *	0.630 ± 0.017 *
PE(38:4)	1.221 ± 0.051	1.159 ± 0.032	1.007 ± 0.023 *	1.111 ± 0.036
PE(P-38:5)	0.988 ± 0.027	0.920 ± 0.014	0.879 ± 0.016 *	0.985 ± 0.026
PE(P-40:7)	0.434 ± 0.019	0.408 ± 0.009	0.343 ± 0.009 *	0.456 ± 0.008
PE(P-40:6)	0.698 ± 0.015	0.683 ± 0.009	0.612 ± 0.011 *	0.728 ± 0.014
PE(P-40:5)	0.583 ± 0.020	0.568 ± 0.010	0.493 ± 0.013 *	0.625 ± 0.014
PI(34:1)	0.313 ± 0.012	0.336 ± 0.019	0.520 ± 0.026 *	0.317 ± 0.013
PI(36:1)	0.268 ± 0.008	0.282 ± 0.012	0.428 ± 0.015 *	0.349 ± 0.010 *
PI(38:5)	0.485 ± 0.012	0.476 ± 0.006	0.424 ± 0.007 *	0.438 ± 0.007 *
PI(38:4)	4.089 ± 0.120	4.040 ± 0.078	3.612 ± 0.076 *	4.189 ± 0.088
PS(38:4)	0.196 ± 0.004	0.177 ± 0.007	0.138 ± 0.003 *	0.218 ± 0.006 *
PG(40:7)	0.164 ± 0.006	0.155 ± 0.009	0.129 ± 0.007 *	0.156 ± 0.012
PG(44:12)	0.056 ± 0.003	0.050 ± 0.002	0.024 ± 0.002 *	0.045 ± 0.004
PG(44:11)	0.020 ± 0.002	0.018 ± 0.001	0.005 ± 0.001 *	0.018 ± 0.002
CL(72:6)	0.196 ± 0.008	0.181 ± 0.003	0.165 ± 0.003 *	0.186 ± 0.004
CL(72:5)	0.133 ± 0.004	0.124 ± 0.004	0.113 ± 0.002 *	0.129 ± 0.004
**Lysophospholipids**				
LPC(16:0)	0.085 ± 0.007	0.091 ± 0.010	0.118 ± 0.011 *	0.083 ± 0.008
LPC(18:0)	0.029 ± 0.003	0.034 ± 0.004	0.042 ± 0.002 *	0.029 ± 0.002
LPI(16:0)	0.010 ± 0.000	0.012 ± 0.001 *	0.022 ± 0.001 *	0.013 ± 0.001
LPI(18:0)	0.062 ± 0.002	0.074 ± 0.003 *	0.123 ± 0.005 *	0.101 ± 0.010 *
**Sphingolipids**				
SM(34:1:2)	6.201 ± 0.069	6.260 ± 0.077	6.305 ± 0.079	6.759 ± 0.071 *
SM(34:0:2)	0.082 ± 0.004	0.087 ± 0.003	0.074 ± 0.005	0.099 ± 0.004 *
Cer(34:1:2)	0.097 ± 0.004	0.090 ± 0.002	0.105 ± 0.002	0.142 ± 0.003 *
Cer(42:2:2)	0.065 ± 0.002	0.073 ± 0.002 *	0.090 ± 0.002 *	0.090 ± 0.002 *
HexCer(34:1:2)	0.155 ± 0.006	0.156 ± 0.006	0.145 ± 0.004	0.203 ± 0.005 *
HexCer(42:2:2)	0.049 ± 0.002	0.049 ± 0.002	0.045 ± 0.001	0.061 ± 0.002 *

Data are presented as mol% of membrane lipids and are expressed as mean ± SEM, *n* = 8. Student’s *t*-tests were performed for pairwise multiple comparisons; significance was accepted for *p* < 0.025 (*) corresponding to a false discovery rate q < 0.05. PC and PC-O, diacyl and alkyl-acyl phosphatidylcholine; PE and PE-P, diacyl and alkenyl-acyl phosphatidylethanolamine; PI, phosphatidylinositol; PS, phosphatidylserine; PG, phosphatidylglycerol; CL, cardiolipin; LPC, LPI, the corresponding lysolipids; SM, sphingomyelin; Cer, ceramide; HexCer, hexosylceramide.

## Data Availability

Raw RNA-seq data have been deposited in the NCBI Gene Expression Omnibus (GEO; https://www.ncbi.nlm.nih.gov/geo/) (accessed on 18 May 2022) under accession number GSE199531.

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
