# Peer review of "Distinct Cellular Tools of Mild Hyperthermia-Induced Acquired Stress Tolerance in Chinese Hamster Ovary Cells"

_biomedicines, 2022, doi:10.3390/biomedicines10051172_

Round 1
Reviewer 1 Report
Major issues:
- For systemic exploration, the authors should perform tests in more than one cell to make the data convincing.
- The stress usual induces translation of existing mRNA at first instead of initiate transcriptional alteration. Accordingly, the authors should screen proteomic changes to make the story completed.
- The author should provide associated "phenotype" changes instead of just screening some signals and metabolites.
Author Response
Answers to Reviewer 1
Point 1: For systemic exploration, the authors should perform tests in more than one cell to make the data convincing.
Response 1:
We agree with the reviewer that further studies are needed to draw a more generalized conclusion. We chose the CHO cell line because this is a well characterized cellular model and at the same time an important industrial target. To avoid generalization, the manuscript has been carefully revised (especially in the discussion and conclusions) to highlight cell specificity of the experimental data and the need for more systematic studies in the future. However, we beleive that our study with CHO (and earlier with MEF) cells is an important example which questions the general view that acquired thermotolerance and Hsp production goes side by side in all animal cells.
Point 2: The stress usual induces translation of existing mRNA at first instead of initiate transcriptional alteration. Accordingly, the authors should screen proteomic changes to make the story completed.
Response 2:
Protein translation could indeed be initiated from preexisting mRNAs, however the detection of these subtle changes in protein levels could be challenging. A recent high-throughput quantitative proteomics and targeted mRNA quantification revealed that a moderate heat treatment of 4 h at 41 °C caused only a minor across-the board mass loss in housekeeping proteins which was matched by a mass gain in a few Hsps, predominantly cytosolic (Finka et al., Cell Stress and Chaperones, 2015, 20, 605-620). However, our Western blot experiments on the unchanged levels of different Hsps (Hsp25, Hsp70) and the house keeping GAPDH in this study, and in our previous study Hsp60, Hsp90, GRP78 (Peksel et al, 2017, Sci Rep, 7, 15643), suggest that during the initial few hours of mild heat the perturbation in the protein production is negligible regardless of the preexisting mRNAs (current study). Previous investigations on avian (chicken reticulocytes) and mammalian (Chinese hamster fibroblast) cells also indicated that a mild temperature treatment up to 41 °C did not affect protein production (Li et al. 1982, P.N.A.S, 79, 3218–3222; Morimoto et al. 1984, J. Cell Biol., 99, 1316–1323). Taking the advice of the reviewer, we have included a new paragraph to discuss this subject in more detail in the manuscript. The use of proteomics for the systematic study of mild heat-induced global change in the cellular proteome is underway in our laboratory.
Point 3: The author should provide associated "phenotype" changes instead of just screening some signals and metabolites.
Response 3:
The observed phenotypic change associated with mild stress treatment was the increase in heat and membrane perturbation stress resistance. We beleive that, based on literature and on our own previous experimental data, both Hsps and membrane lipid reorganization could explain the increased stress resistance and thereby the existence of “phenotypic” change. (Li et al. 1982, Peksel et al, 2017).
Reviewer 2 Report
Ref: biomedicines-1701648
Title: Distinct cellular tools of mild hyperthermia-induced acquired stress tolerance in Chinese Hamster Ovary cells
Recommendation: MAJOR REVISION / REJECT
In this study, the Authors aimed to verify an impact of fever-type mild heat on lipid rearrangements and membrane stress. To verify the hypothesis, the Authors used CHO cell line and several methods including determination of ATP, measurement of the mitochondrial membrane potential, assessment of lipid peroxidation as well as lipidomics RNAseq experiments. The main strength of this paper are RNAseq lipidomics experiments. Nevertheless, the paper has several flaws which are listed below.
Comments:
- The main objection applies to the use of the CHO line. The Authors do not mention why they chose CHO cells to verify effects of hyperthermia. What is the meaning of these results e.g. for clinic, for basic biology etc. It is known that cell lines react very differently from ‘wild-type’ / natural / primary cells.
- The section 2.1. is very poor. It makes impossible to recreate / repeat the experiment by another group. Please add information about cells density,timeframe of cell culture, medium changing schedule etc.
- Part 2.2 – there is no information about antibodies dilutions.
- Please add the information about JC-1 data presenting. Did Authors count ratio between green and red fluorescence?
- Please include in the main manuscript the effects of hyperthermia on JC-1 parameter.
- There should be another paragraph about Statistic analyses with numbers of replicates, ‘n’ number etc.
- 1a and 1b are incomprehensible. I do not see differences between 40 and 42.5 C.
- A lot of interesting data are presented as a supplement, e.g., S5.
Author Response
Point 1: The main objection applies to the use of the CHO line. The Authors do not mention why they chose CHO cells to verify effects of hyperthermia. What is the meaning of these results e.g. for clinic, for basic biology etc. It is known that cell lines react very differently from ‘wild-type’ / natural / primary cells.
Response 1:
The use of primary cells to study mild stress response could be very challenging because of the additional stress caused by the cell isolation itself. We chose the CHO cell line because this is a well characterized cellular model and at the same time an important industrial target. Based on our initial observations further studies could reveal more generalized conclusions. Our study gives an important example which questions the general view that acquired thermotolerance and Hsp production goes side by side.
Point 2: The section 2.1. is very poor. It makes impossible to recreate / repeat the experiment by another group. Please add information about cells density,timeframe of cell culture, medium changing schedule etc.
Response 2:
We have corrected section 2.1 as the reviewer suggested.
Point 3: Part 2.2 – there is no information about antibodies dilutions.
Response 3:
We have included the dilutions in section 2.2 as the reviewer suggested.
Point 4: Please add the information about JC-1 data presenting. Did Authors count ratio between green and red fluorescence?
Response 4:
Yes, we measured the ratio of green and red fluorescence. We improved the description of the experiment in section 2.5 accordingly.
Point 5: Please include in the main manuscript the effects of hyperthermia on JC-1 parameter.
Response 5:
Following the heat shock, JC-1 was added and fluorescence was measured at 23 °C to avoid the effect of temperature on JC-1 fluorescence. We included this information in section 2.5.
Point 6: There should be another paragraph about Statistic analyses with numbers of replicates, ‘n’ number etc.
Response 6:
Following the reviewer’s advice, statistical analyses were moved to a new, separate section (2.9).
Point 7: 1a and 1b are incomprehensible. I do not see differences between 40 and 42.5 C.
Response 7:
Figure 1a indicates that there is no Hsp25 induction either at 40 °C (empty black circle) or at 42.5 °C (empty red square) if measured right after 1 h heat treatment. However, figure 1b shows that if 1 h heat treatment at the above temperatures is followed by 6 h recovery period at 37 °C, the two distinct doses of heat stress result in a remarkable difference in Hsp25 synthesis. We have highlighted the single 42.5 °C points on figure 1a and 1b with red color.
Point 8: A lot of interesting data are presented as a supplement, e.g., S5.
Response 8:
Following the advice of the reviewer, one figure (S5b) was moved from the supplement to the manuscript as a new figure panel (fig 3a).
Round 2
Reviewer 1 Report
There is no improvement made in the revision.
Author Response
We have revised the manuscript based on the suggestions of the editors.
Reviewer 2 Report
The authors made some corrections to the manuscript.
Nevertheless, in my opinion, the paper does not seem to be sufficient for a publication in Biomolecules - IF 6.081.
Author Response

(The authors gave the same response as above.)
